# A Pilot Study of the Psychosocial Impact of Low-Cost Assistive Technology for Sexual Functioning in People with Acquired Brain Injury

**DOI:** 10.3390/ijerph18073765

**Published:** 2021-04-04

**Authors:** Estíbaliz Jiménez, Feliciano Ordóñez

**Affiliations:** Facultad Padre Ossó, Universidad de Oviedo, 33003 Oviedo, Asturias, Spain; feliciano@facultadpadreosso.es

**Keywords:** assistive technology, sexuality, outcomes, acquired brain injury

## Abstract

People with acquired brain injury (ABI) face limitations when performing activities of daily living, including sexuality. Despite the common use among this group of assistive technology to compensate for or neutralize the limitations deriving from their condition, there is very little literature on outcome measures in assistive technology for sexual functioning. The aim of this study was to explore the psychosocial impact of the use of low-cost assistive technology in people with ABI. The sample was made up of 18 users: 15 men and 3 women diagnosed with ABI. The PLISSIT model was used, as well as the Psychosocial Impact of Assistive Device Scale—PIADS as an assessment tool. Three types of low-cost assistive technology were developed: seat cushions, bed equipment, and back supports. All three types of AT obtained positive scores on the PIADS total scale and its three subscales: competence, adaptability, and self-esteem. Although the results of this study are positive, more research into outcome measures for products to improve sexual functioning in people with ABI is required.

## 1. Introduction

A common intervention strategy among disabled people to improve, mitigate, or compensate for their disability is the use of assistive technology (henceforth AT) [1,2,3,4,5,6]. A number of systematic reviews have analyzed customized AT for different conditions, such as dementia [7], learning disability [8], multiple sclerosis [9], rheumatoid arthritis [10], and cerebral palsy [11]. Increasingly, studies are focusing on AT in relation to the activities of daily living that it enables, including AT for improving mobility [12,13,14,15].

Assistive technology may be classified in different ways: In this study, we use the classification proposed by Cook and Polgar [1], which divides AT into high technology [difficult to obtain, electronic] and low technology [easy to make and obtain]. Low technology products are made by many clinicians, including occupational therapists, as customized AT sometimes fails to fulfil users’ needs [16], is expensive [17], or delivers poor performance [16]. Occupational therapists and direct care professionals use devices of this kind for various reasons: (1) the high cost of some customized AT, which can be reproduced at a lower cost; (2) the absence of AT that meets the user’s needs and preferences on the market; (3) the need to adapt an existing form of AT to make it suitable for the user. The scientific literature contains studies on low-cost AT, such as AT for mobility [18,19,20,21], AT for computer access [22,23], AT for educational activities [24], audio prescription [25], AT for blind people, e.g., creating graphics [26], wheelchair accessories [27], gloves for hand therapy [28], and, in recent years, 3D printers for making prostheses [29,30] and orthoses [31], which are increasingly widely used. The do-it-yourself movement has also influenced disabled people and their relatives to make their own low-cost AT [32,33] and users gather to share ideas at events such as the low-cost gatherings in Spain [34] or on websites like Thingiverse [35] or even Pinterest [36].

Information on the use of AT for sexual functioning is very scarce in scientific literature (this must be understood as healthy sexual functioning characterized by a lack of pain or discomfort during sexual activity) [37]. Previous literature on sexual functioning address intervention tools such as sexual education, which is an adequate strategy to provide direct information on healthy, satisfactory, and responsible sexuality. However, it is necessary to continue researching intervention and assessment procedures to demonstrate the effectiveness of the interventions, in addition to it being necessary to develop intervention programs adapted to individual needs. Previous studies have developed intervention programs on stroke; for example, the study of Song and colleagues [38], obtained positive results in terms of the continuation of sexual activities and previous sexual roles on stroke patients and their spouses. This study concluded that these kinds of program could be useful. However, there are enough evidence-based interventions to address sexuality and sexual functioning [39,40].This lack of literature on sexual functioning can also be seen in the outcome measures and AT. We find that studies have centered around creating body parts such as genitals and educating people with visual impairments in using male and female condoms [41,42]. However, there are various publications that disclose the deterioration of sexual functioning and the presence of psychological distress [43] as well as the relationship between sexual satisfaction and better quality of life [44].

Disabled people often experience difficulties and limitations in sexuality and sexual functioning, including people with an acquired brain injury (henceforth, ABI). Sexual dysfunction is common after acquired brain injury [45,46,47], which manifests in problems such as erectile dysfunction (ED) and psychological disease [48]. Many ABI are keen to discuss sexual functioning during the rehabilitation process [49,50]. Prior et al. (2019) [51] indicate that 30% of these patients wish to receive information on the topic but only 8.2% actually do. Other authors such as McCluskey [52] show that one of the barriers to intervention in this group’s sexual functioning is the lack of existing knowledge on the subject. There is no scientific evidence to justify the best intervention strategies for sexual functioning [53] or the use of AT among this group, although AT such as splints [54] is commonly used for mobility [13], including upper limb mobility [55]. However, no previous studies have used AT for sexual functioning as an intervention strategy. These are covered by ISO 9999:2017 under the category of AT for self-care activities and participation in self-care (ISO Code 09) and divided into: assistive products for sexual activities (devices for training and assisting during sexual activities, back supports, seat cushions, and beds and bed equipment), assistive products for erection, vibrators and massage devices for sexual activity, and finally, assistive products for sexual rehabilitation.

In the present study, the PLISSIT model was used [56]; this encourages health professionals to provide four levels of intervention that identify their role in assessment and intervention with disabled people to improve their sexual wellbeing. PLISSIT stands for permission [P], limited information [LI], specific suggestions [SS], and intensive therapy [IT]. Taylor and Davis (2007) [57] in their development of Ex-PLISSIT state that “each stage in the Ex-PLISSIT model has Permission-giving at its core All interventions should begin with Permission, and it is essential that this Permission-giving is explicit, giving individuals the opportunity to ask questions or voice their concerns” (p. 136). This phase of permission and that the participant asks questions about the design and use of the AT is essential for the intervention, in other words, the opinion was considered during device selection, assessment, and design process. Using this model and derived from the assessment of the participants, different difficulties in motor control from ABI were identified. Therefore, the ATs that were examined were seat cushion, bed equipment, and back support consistent with the ISO standard and which aimed to improve the positioning of the users. This model was chosen because it has previously been used in this population [58] and other models follow the same general principles. AT can be included in the third step of the “specific suggestions” model, since this allows taking into account your sexual needs in order to be able to perform sexual activities with your partner, which improves sexual satisfaction, using the AT that allows the position sexual rehabilitation in those users with motor and sensory limitations [38,59]. It should be noted in the present study that sexuality does not refer to sexual orientation.

Therefore, the objective of this study was to analyze the psychosocial impact of the use of AT for sexual functioning in people with ABI.

## 2. Materials and Methods

### 2.1. Study Design

An exploratory, comparative design was adopted to fulfil the study objective.

### 2.2. Participants

The sample comprised 18 participants, 15 men (83.3%) and 3 women (16.7%), with an average age of 59.17 (*SD* = 12.16). The sociodemographic profile of the participants can be seen in Table 1. With regard to the inclusion criteria for the study, participants had to have been diagnosed with ABI, be aged over 18 years old, and score 28 points on the Functional Assessment Measure (FAM scale and 91 points in total on the Functional Independence Measure (FIM) + (FAM) scale. Meanwhile, the exclusion criteria were: being unable to understand verbal language, being legally incapacitated, not being in a stable relationship prior to their ABI, and having an aggressive or apathetic behavioral disorder. A total of 27 individuals wanted low-cost AT, but 9 of them did not wish to participate in the study. The study participants correspond to an association of people with neurological involvement (In addition, 4 private clinics were contacted to care for people with neurological pathology), where the objective of the study and its procedure were explained to them and if they wished to participate. A total of 27 participants wanted to participate in the intervention but 9 of them did not want to participate in the study. Participants were originally from three autonomous communities but all lived in the Principality of Asturias at the time of the study. A meeting was held with the occupational therapist of the center and he was the person who made the contact to participate in the study (taking into account the inclusion/exclusion criteria). The study was performed between July 2019 and February 2020. All participants received detailed information about the nature and characteristics of the study and signed an informed consent form based on the model from the Bioethics Committee of the Principality of Asturias.

### 2.3. Instruments

The research group developed an ad hoc sociodemographic questionnaire to collect data on personal aspects.

The Psychosocial Impact of Assistive Devices Scale (PIADS) [60] is a 26-item self-report survey which assesses the functional independence, wellbeing, and quality of life linked to the use of an assistive device. The scores in the PIADS are divided into three subscales: (1) competence, which reflects perceptions of functional capacity, independence, and performance; (2) adaptability, which reflects inclination or motivation to participate socially and take risks; and (3) self-esteem, which reflects confidence, self-esteem, and emotional wellbeing. Users must respond to all items using a 7-point scale which extends from −3 (it has reduced) to +3 (it has increased). The middle point, zero, would indicate that no impact or change has been perceived to result from the use of the device. In the present study, the version of PIADS translated into Spanish was used [61].

### 2.4. Procedure

The data collection procedure was based on the application of the researchers’ own sociodemographic questionnaire and the PIADS scale.

This scale has already been used in previous studies in a population with ABI using different AT [62,63,64]. This scale has excellent psychometric properties in terms of: internal consistency, measurement error, structural validity, criterion validity and responsiveness, and good properties such as reliability and content validity [65]. All interviews in centers and clinics were carried out by an occupational therapist with experience in AT and people with acquired brain injury. Data collection was carried out by means of individual interviews lasting from 40 to 90 min, depending on the characteristics of the participants. Assessment of the user’s needs and creation of the AT ranged from 3 to 6 h depending on the number of times that it had to be re-evaluated to suit the user. The most commonly used materials were foams of different densities and types, including viscoelastic foam and thermoplastic foam in sheets and granules, as well as waterproof fabrics for easy cleaning. Data analysis was carried out using SPSS version 24. All participants signed an informed consent form and the study was approved by the Bioethics Committee.

## 3. Results

### Data Analysis

Due to the type of dependent variables used in the study, that configure the psychosocial variables of the general objective—the ordinal PIADS scale and the sociodemographic variables, which include nominal and ordinal variables—and the number of cases studied, we first conducted a Kruskal–Wallis H test for non-parametric comparison of k-groups. In the case of our study, the independent variable is the type of AT, which segments the sample into three independent groups: G1: Seat cushions 38.9% (*n* = 7); G2: Bed equipment 33.3% (*n* = 6); and G3: Back supports 27.8% (*n* = 5).

This first analysis revealed the variables that display significant differences between the study groups; once detected, the groups could be compared to identify the assistive product and the variables that maintain these differences. To do this, possible differences between the groups for the detected variables were compared using the Mann–Whitney U test, taking two groups at a time.

The following results were obtained for the sociodemographic variables derived from the ad hoc questionnaire: the average time since ABI diagnosis was 3.5 years, with a deviation of almost one year (0.929) and a range of 1 to 4 years. The degree of disability ranged from 33 to 87, with an average of 55 (SD = 20.7). With regard to the degree of dependence, the sample broke down as follows: Degree I, 5.6%; Degree II, 22.2%; Degree III, 5.6%, with a total of 66.7% of participants responding to this question. As for the degree of ambulation, 44.4% of participants were community ambulators, 11.1% were household ambulators, and 22.2% were nonfunctional ambulators and non-ambulators, respectively. Secondary diagnoses alongside ABI were: AHT 11.1%, diabetes 11.1%, sleep apnea 5.6%, epilepsy 5.6%, depression 11.1%, alcoholism 5.6%, obesity 5.6%, and smoking 5.6%. In terms of the different types of treatment received, 88.9% of participants had received occupational therapy and physiotherapy, 38.9% had received speech therapy, 22.2% had received psychological treatment, and 5.6% had received alternative therapies.

Table 2 shows the average scores by item and by subscale for all study participants.

The main objective of the study was to ascertain the perceived psychosocial impact of the use of low-cost AT and the score obtained for the PIADS total scale was 1.53 (SD = 0.69). The scores for the three subscales were: competence subscale 1.49 (SD = 0.59); adaptability subscale 1.36 (SD = 0.75); and self-esteem subscale 1.73 (SD = 1.08).

Table 3 shows the results of the significant variables from the Kruskal–Wallis test to compare k independent samples, taking the type of AT as the group distribution variable and the items of the PIADS as sociodemographic variables. As the table shows, only one sociodemographic variable appears—the degree of ambulation—along with six items from the PIADS and two of its subscales: self-esteem and competence.

In the items from the PIADS, the highest range belongs to bed equipment, whereas in the degree of ambulation variable, it belonged to seat cushions.

Table 4 shows the results of the two-by-two comparative analysis (Mann–Whitney U) of the variables that obtained significant values in the Kruskal–Wallis H test. The distribution variable used in the comparative analysis was the type of product used in sexual activity: G1 Seat cushions, G2 Bed accessories, and G3 Back support.

The results show that the group using seat cushions and the group using bed equipment obtained significant differences in the ambulation variable, with the highest average value found in the seat cushions group (average range G1: 9.29 vs. G2: 4.33). Meanwhile, in the items from the PIADS, the highest values for all variables were found in the bed equipment group: productivity (average range G1: 4.50 vs. G2: 9.92), confidence (average range G1: 5.29 vs. G2: 9.00), sense of power (average range G1: 5.29 vs. G2: 9.00), sense of control (average range G1: 5.29 vs. G2: 9.00), and competence subscale (average range G1: 4.64 vs. G2: 9.75). These differentiating variables appear to be more closely related to the bed equipment than to the seat cushions.

With regard to the quality of life and self-esteem subscale variables, their significance level depends on the differences in values for the other AT, not the bed equipment group. More specifically, quality of life (average range G1: 8.71 vs. G3: 3.40) and the self-esteem subscale (average range G1 = 8.29 vs. G3 = 4.00) obtain significant values in the comparison between seat cushions and back supports. Unlike the previous variables, these variables appear to be more closely linked to the use of seat cushions.

Finally, the results of the comparison between the bed equipment and back support groups obtained significant differences in all variable s linked to the PIADS, with a higher average range in the values for the bed equipment group. The reliability analysis produced excellent results: the PIADS total scale obtained McDonald’s ω 0.929 and Cronbach’s α 0.902. The scores for the subscales were as follows: competence subscale: McDonald’s ω 0.810 and Cronbach’s α 0.763; adaptability subscale: McDonald’s ω 0.835 and Cronbach’s α 0.800; and self-esteem subscale: McDonald’s ω 0.792 and Cronbach’s α 0.578.

## 4. Discussion

Research into ABI is a particularly active line of inquiry due to the high prevalence of the condition. A number of previous studies have used the PLISSIT model to explore sexual functioning in disabled people, such as ostomized people, people with cancer, and people with spinal cord injury [66,67], but there has been no research using this model in relation to low-cost AT for sexual functioning in people with ABI. Although sexuality is recognized as an important part of the rehabilitation process for people with ABI and patients often wish to receive information about this subject, only a small percentage actually do so [51]. Sexuality continues to be viewed negatively in many spheres and disabled people suffer even greater stigma, as several prior studies have shown [68]. In this study, 9 of the 27 initial subjects withdrew their participation because they viewed sexuality as a taboo topic.

With regard to the study objective, the psychosocial impact of the use of different low-cost AT was positive on the PIADS total scale and its three subscales for 17 participants. One participant obtained a score of −0.07 on the PIADS total scale (he had been diagnosed with depression) and another participant obtained a positive score that was close to 0 (0.07); both of them were using back supports, which obtained a lower score than the other AT groups and scored the lowest on the self-esteem subscale. The group that used bed equipment obtained higher average scores than the other groups in the items relating to the PIADS.

The group using seat cushions was the only one to display significant differences in the sociodemographic variable of ambulation, as well as obtaining better results than the group using back supports. Previous research has concluded that community ambulation is a significant outcome following ABI [69] and the loss of independent community ambulation is one of the most disabling consequences of ABI [70]. The PIADS has been used in studies of a wide range of AT, including research on the psychosocial impact of wheelchair use [19], AT for mobility in people with neurological disorders [18], AT for urinary incontinence, electronic health services [71], and voice recognition [72]. Despite this, it has not been used to study sexual functioning and although the results obtained in this study cannot be extrapolated, they indicate that the scale can be used to evaluate the impact of this type of AT. In previous studies using the PIADS, positive scores were obtained for both the total scale and the three subscales. In this study, the competence and self-esteem subscales were significant, echoing the findings of Yachnin et al. [73] who studied the use of technology-assisted toilets in people with ABI. However, the self-esteem subscale scored lower in other studies of people with neurological disorders, including research on neuromuscular disorders and the use of wheelchairs [14]. With regard to the internal consistency of the PIADS [60], the original study of the scale indicates a Cronbach’s α of 0.95 for the total scale, while the scores for the subscales were as follows: competence 0.92, adaptability 0.88, and self-esteem 0.87. Similar values were obtained in this study, although the scores were somewhat lower than those in the original study; the self-esteem scale obtained the lowest score. There is no clear indication as to why this subscale has a lower internal consistency, but it may be due to the variability of the responses. Further research on self-esteem in relation to sexual activity in people with ABI is required.

The PIADS is conceptually compatible with the International Classification of Functioning, Disability and Health (henceforth ICF) [74]. The ICF includes sexual functioning within the ‘body function’ dimension. Earlier studies have argued that the ICF can guide professionals and suppliers in their decision-making with regard to AT and that it can be used to evaluate outcome measures based on evidence [75]. Many authors have discussed the potential use of the ICF in measures of results [76] and there are also many instruments for the selection of AT that include the ICF components [77].

The PIADS can be a useful tool for evaluating this type of AT, but more exhaustive research is needed to inform clinicians of the best intervention strategies for use with this group [53], as well as ways of continuing these interventions in outpatient settings [71] using AT among other measures.

Among the main limitations of this study are the small sample size and the ad hoc development of the AT, as there was no existing scientific literature to provide guidance on products for sexual functioning, although there are several websites selling this type of AT for disabled people. However, these websites tend to focus primarily on people with spinal cord injury; there is also a greater number of studies exploring sexual rehabilitation in these patients, including sexuality after spinal cord injury (SCI) [78], and websites that are more general in scope [79]. It is important that research in this area continues to improve knowledge of outcome measures and assistive products for sexuality in people with ABI and other physical disabilities.

## 5. Conclusions

AT is a common intervention strategy among disabled people to mitigate and/or compensate for limitations to their occupational performance. (ATs are classified in the ICF in environmental factors in chapter 1 and are defined “as any product, instrument, equipment or technology adapted or specially designed for improving the functioning of a disabled people” (p. 180) [74]). Among AT as a whole, low-cost assistive products are of particular interest. Despite their frequent use among rehabilitation professionals and occupational therapists, there is little scientific literature on the subject. Although the results of this study cannot be extrapolated to other contexts, they offer a foundation for more in-depth studies using AT for sexuality as an intervention strategy among people with ABI and their partners.

## Figures and Tables

**Table 1 ijerph-18-03765-t001:** Sociodemographic profile of study participants (*n* = 18).

Variable	*N* (%)
Marital status	
In a relationship	3 (16.7)
Married	15 (83.3)
Autonomous community	
Asturias	4 (22.2)
Castile and León	9 (50)
Extremadura	1 (5.6)
Unanswered	4 (22.2)
Education	
No education	1 (5.6)
Primary education	4 (22.2)
Secondary education	6 (33.3)
University	5 (27.8)
Unanswered	2 (11.1)
Degree of ambulation	
Community ambulator	8 (44.4)
Household Ambulators	2 (11.1)
Nonfunctional ambulator	4 (22.2)
Non-ambulator	4 (22.2)
Treatment received	
Occupational therapy	16 (88.9)
Physiotherapy	16 (88.9)
Speech therapy	7 (38.9)
Psychotherapy	4 (22.2)

**Table 2 ijerph-18-03765-t002:** Average scores (SD) per item on the competence, adaptability, and self-esteem subscales of the PIADS (Psychosocial Impact of Assistive Devices Scale).

Competence (12 Items)	Adaptability (6 Items)	Self-Esteem (8 Items)
Competence; *M* = 1.67 (*SD* = 1.3)	Wellbeing; *M* = 1.72 (*SD* = 0.89)	Happiness; *M* = 2.22 (*SD* = 1.26)
Independence; *M* = 1.89 (*SD* = 0.96)	Willingness to take chances; *M* = 1.61 (*SD* = 0.92)	Self−esteem; *M* = 2.39 (*SD* = 0.98)
Adequacy; *M* = 1.61 (*SD* = 1.2)	Ability to participate; *M* = 1.67 (*SD* = 0.97)	Security; *M* = 1.89 (*SD* = 1.32)
Confusion *; *M* = −0.89	Eagerness to try new things; *M* = 1.22 (*SD* = 1.16)	Frustration; *M* = −1.61 (*SD* = 1.29)
(*SD* = 1.37)	Ability to adapt to the activities of daily living; *M* = 1.44 (*SD* = 0.92)	Confidence; *M* = 2.17 (*SD* = 1.1)
Efficiency; *M* = −1.39	Ability to take advantage of opportunities; *M* = 1.33 (*SD* = 0.97)	Sense of power; *M* = 1.72 (*SD* = 1.36)
(*SD* = 1.24)		Sense of control; *M* = 1.72 (*SD* = 1.34)
Productivity; *M* = 83		Embarrassment *; *M* = −1.39 (*SD* = 1.54)
(*SD* = 1)		
Usefulness; *M* = 1.33		
(*SD* = 1.45)		
Expertise; *M* = 1		
(*SD* = 0.97)		
Skillfulness; *M* = 1.39 (*SD* = 1.3)		
Capability; *M* = 2.22 (*SD* = 0.94)		
Quality of life; *M* = 2.28 (*SD* = 0.83)		
Performance; *M* = 1.5 (*SD* = 0.98)		

* negative items.

**Table 3 ijerph-18-03765-t003:** Results of the significant variables in the initial Kruskal–Wallis H test with type of Assistiv Technology as the independent variable, and the PIADS as sociodemographic variables.

Variable	AT	Average Range	Chi-Square (Gl.)	Sig./*p*
Degree of ambulation	Seat cushions	13.07		
	Bed equipment	5.83		
	Back supports	8.90	6.759 (2)	0.034
Self-esteem				
	Seat cushions	10.14		
	Bed equipment	12.50		
	Back supports	5.00	7.930 (2)	0.019
Productivity				
	Seat cushions	7.71		
	Bed equipment	14.92		
	Back supports	5.50	12.087 (2)	0.002
Confidence				
	Seat cushions	8.93		
	Bed equipment	13.50		
	Back supports	5.50	7.625 (2)	0.022
Quality of life				
	Seat cushions	11.00		
	Bed equipment	11.67		
	Back supports	4.80	6.408 (2)	0.041
Sense of power				
	Seat cushions	9.00		
	Bed equipment	14.00		
	Back supports	4.80	9.702 (2)	0.008
Sense of control				
	Seat cushions	9.00		
	Bed equipment	14.00		
Competence subscale	Back supports	4.80	9.702 (2)	0.008
	Seat cushions	8.71		
Self-esteem subscale	Bed equipment	14.75		
	Back supports	4.30	10.832 (2)	0.004
	Seat cushions	9.79		
	Bed equipment	14.25		
	Back supports	3.40	11.345 (2)	0.003

*p*: Significance ≤ 0.05.

**Table 4 ijerph-18-03765-t004:** Significant values from the two-by-two comparison of the study groups using the Mann–Whitney U test.

Groups	Bed Equipment (G2)	Back Supports (G3)
Seat cushions	Degree of ambulation	Quality of life
(G1)	*p* = 0.014	*p* = 0.031
	Productivity	Self-esteem subscale:
	*p* = 0.008	*p* = 0.012
	Confidence	
	*p* = 0.036	
	Sense of power	
	*p* = 0.036	
	Sense of control	
	*p* = 0.036	
	Competence subscale:	
	*p* = 0.017	
Bed equipment		Self-esteem
(G2)		*p* = 0.011
		Productivity
		*p* = 0.003
		Confidence
		*p* = 0.011
		Quality of life
		*p* = 0.030
		Sense of power
		*p* = 0.002
		Sense of control
		*p* = 0.002
		Competence subscale
		*p* = 0.006
		Self-esteem subscale
		*p* = 0.006

*p*: Significance ≤ 0.05.

## Data Availability

Not applicable.

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
