# Peer review of "A Pilot Study of the Psychosocial Impact of Low-Cost Assistive Technology for Sexual Functioning in People with Acquired Brain Injury"

_ijerph, 2021, doi:10.3390/ijerph18073765_

Round 1

Reviewer 1 Report

This article is of relevant scientific and clinical value. Overall the manuscript is easy to read. Terminology needs to be reviewed. English language and style require minor spell check.

Introduction describes the state of the art in a pertinent way, using significant literature on the topic, including relevant and actual references.

Methods and results needs improvement (please see my notes along the text). After that, discussion needs to be reformulated.

Author Response

Thank you very much for your contributions and suggestions.
You can see the changes made in the attached document

Reviewer 2 Report

The presented manuscript is very interesting and highly relevant. It is worth paying attention to the development of low-cost technologies to improve the daily life of people with acquired brain damage.

On the other hand, I must congratulate the authors for paying attention to sexual aspects. This is something that is often forgotten in rehabilitation processes. 

Here are some comments with the aim of improving the submitted manuscript:
- The first sentence of the manuscript needs to be underpinned with bibliographical references.
- In the second sentence the authors mix pathologies with disability and communication systems. Order is needed.
- It would be interesting to define what the authors mean by "sexual function".
- It is advisable to separate the objective in a single paragraph.
- In the methodology, it is advisable to expand on the information of the study. Citing, in addition, others of similar characteristics.
- On the other hand, it would be interesting to know why these sexual function support products were chosen and not others. 
- It is not at all clear how the sample was obtained; could this information be expanded?
- It is not clear how the authors use the PLISSIT method. This needs to be explored further.
- Line 246 et seq: This paragraph is not discussing the results. Could this paragraph be incorporated into the introduction?

Author Response

(The authors gave the same response as above.)

Round 2

Reviewer 1 Report

The majority of the comments were achieved. Please see attached document.

Author Response

In the attached document are the responses to the reviewer's written comments. Thanks for your time.

REVIEWER 1

Page 4 line 136

The degree of dependency is assessed with a tool called the Dependency Assessment Scale. The objective is to assess the level of performance of the set of tasks considered, as well as the frequency with which  support is needed to carry them out. They are carried out by public service workers.

Table 4.

We believed it convenient to analyze the items detected as relevant in the previous global analysis (Kruskal-Wallis) in groups two by two, in order to specifically identify those variables that were truly significant by type of AT, discarding joint values. In addition to eliminating possible errors when taking multiple analyzes.

Page 6 line 139

Regarding the use of ranges, it has no other meaning than the use of non-parametric analysis, in particular the Kruskal-Wallis comparison test for K samples as it is a non-parametric test that analyzes the hypotheses in ranges, uses the median and the frequency of the data. While parametric tests use hypotheses based on numerical data. All this is due to the fact that the distribution of the variables is non-normal. The procedure has shown good properties in numerous non-normal distributions both in control of Type I Error and in power or statistical sensitivity. (Oliver Rodríguez. González Álvarez and Rosel Remirez, 2009).

Oliver Rodríguez, J.C. ,González Álvarez, J. and Rosel Remirez, J.(2009). Análisis no paramétrico de la interacción de dos factores mediante el contraste de rangos alineados. Psicothema, 21 (1). 152-158.

Page 7 line 192

In the instrument section, it was added in this second version that the scale used in the present study corresponds to the Spanish version.

Page 8 line 198

We believed it convenient to analyze the items detected as relevant in the previous global analysis (Kruskal-Wallis) in groups two by two, in order to specifically identify those variables that were truly significant by AT, discarding joint values. In addition to eliminating possible errors when taking multiple analyzes

Page 8 line 202

The phrase was already removed in the second versión after the first revisión of the article.

Page 9 line 247

Has been included “Many authors have discussed the potential use of the ICF in measures of results (Arthanat & Lenker, 2004; Jutai, Fuhrer, Demers, Scherer, & DeRuyter, 2005; Scherer & Jutai, 2007) and there are also many instruments for the selection of AT that include the ICF components (Scherer & Sax, 2005). On the one hand, the CIF pays attention to functioning and disability and on the other to contextual factors (Jutai, Fuhrer, Demers, Scherer, & DeRuyter, 2005; Scherer, Jutai, Fuhrer, Scherer & Glueckauf, 2005).”

Page 9 line 265

Has been included “Included in the ICF in part one functioning and disability in the activities and participation dimensions).”

Page 10 line 273

Has been included. Estíbaliz Jiménez Arberas: Conceptualization, Investigation, Resources, Data Curation, Writing - Review & Editing. Feliciano Ordóñez Fernández: Methodology, Formal analysis, Data Curation, Writing - Original Draft, Writing

Reviewer 2 Report

Unfortunately, the authors of the manuscript have not heeded some of the recommendations made in the first review.

In fact, they have not made any changes to the introduction when there is clearly room for improvement.
- The first sentence of the manuscript needs to be underpinned with bibliographical references.
- In the second sentence the authors mix pathologies with disability and communication systems. Order is needed.
- It would be interesting to define what the authors mean by "sexual function".
- In the methodology, it is advisable to expand on the information of the study. Citing, in addition, others of similar characteristics.
- On the other hand, it would be interesting to know why these sexual function support products were chosen and not others. 
- It is not at all clear how the sample was obtained; could this information be expanded?
- It is not clear how the authors use the PLISSIT method. This needs to be explored further.

Author Response

In the attached document are the responses to the reviewer's written comments. Thanks for your time.

REVIEWER 2

- The first sentence of the manuscript needs to be underpinned with bibliographical references.

It has been expanded with a greater number of references with articles with evidence

- In the second sentence the authors mix pathologies with disability and communication systems. Order is needed.

It has been modified. We hope it is better understood now

- It would be interesting to define what the authors mean by "sexual function".

Has been included

- In the methodology, it is advisable to expand on the information of the study. Citing, in addition, others of similar characteristics.

Has been included. The PIADS tool has been used in many studies that investigate the impact of AT on occupational performance in a variety of pathologies and products. To see more information about its use you can consult: http://piads.at/

- On the other hand, it would be interesting to know why these sexual function support products were chosen and not others.

 These assistive technologies were chosen because they are the ones included in the ISO standard, in addition to the fact that in different resources these are the ones that are most used in related pathologies such as spinal cord injury (https://www.sexualitysci.org/adaptive-devices)

- It is not at all clear how the sample was obtained; could this information be expanded?

Has been included

- It is not clear how the authors use the PLISSIT method. This needs to be explored further.

Has been included
